# Extracellular Vesicles as Markers of Liver Function: Optimized Workflow for Biomarker Identification in Liver Disease

**DOI:** 10.3390/ijms24119631

**Published:** 2023-06-01

**Authors:** Martha Paluschinski, Sven Loosen, Claus Kordes, Verena Keitel, Anne Kuebart, Timo Brandenburger, David Schöler, Marianne Wammers, Ulf P. Neumann, Tom Luedde, Mirco Castoldi

**Affiliations:** 1Department of Gastroenterology, Hepatology and Infectious Diseases, Medical Faculty, Heinrich-Heine University, 40225 Düsseldorf, Germany; martha.paluschinski@uni-duesseldorf.de (M.P.); sven.loosen@med.uni-duesseldorf.de (S.L.); claus.kordes@uni-duesseldorf.de (C.K.); verena.keitel-anselmino@med.ovgu.de (V.K.); david.schoeler@med.uni-duesseldorf.de (D.S.); marianne.wammers@gmail.com (M.W.); tom.luedde@med.uni-duesseldorf.de (T.L.); 2Department of Anesthesiology, Medical Faculty, Heinrich-Heine University, 40225 Düsseldorf, Germany; anne.kuebartr@med.uni-duesseldorf.de (A.K.); timo.brandenburger@med.uni-duesseldorf.de (T.B.); 3Visceral and Transplant Surgery, University Hospital RWTH Aachen, 52074 Aachen, Germany; uneumann@ukaachen.de

**Keywords:** extracellular vesicles, nanoparticle-tracking analysis, liver diseases, biomarker, microRNA, cytokine

## Abstract

Liver diseases represent a significant global health burden, necessitating the development of reliable biomarkers for early detection, prognosis, and therapeutic monitoring. Extracellular vesicles (EVs) have emerged as promising candidates for liver disease biomarkers due to their unique cargo composition, stability, and accessibility in various biological fluids. In this study, we present an optimized workflow for the identification of EVs-based biomarkers in liver disease, encompassing EVs isolation, characterization, cargo analysis, and biomarker validation. Here we show that the levels of microRNAs miR-10a, miR-21, miR-142-3p, miR-150, and miR-223 were different among EVs isolated from patients with nonalcoholic fatty liver disease and autoimmune hepatitis. In addition, IL2, IL8, and interferon-gamma were found to be increased in EVs isolated from patients with cholangiocarcinoma compared with healthy controls. By implementing this optimized workflow, researchers and clinicians can improve the identification and utilization of EVs-based biomarkers, ultimately enhancing liver disease diagnosis, prognosis, and personalized treatment strategies.

## 1. Introduction

Extracellular vesicles (EVs) are a heterogeneous group of lipid-enclosed nanoparticles secreted by prokaryotic and eukaryotic cells that are mediators of intercellular communication [1]. EVs carry biomolecules, including proteins, metabolites, cytokines, mRNAs, and microRNAs (miRNAs), and have the ability to transport these cargoes to recipient cells over short and long distances. The ability of EVs to be transported by the blood and to deliver their payloads to recipient cells makes them excellent candidates for the delivery of agents in targeted therapies and for monitoring disease-associated biomarkers [2]. Despite these promising potentials, the use of EVs in the clinical setting is still limited due to the lack of standardized EVs classification, isolation protocols, and analysis methods. The three most prominent subtypes of EVs are exosomes (Exo; [3]), microvesicles (MVs), and apoptotic bodies (AP), which have been differentiated based upon their biogenesis, release pathways, size, content, and function [4,5]. While Exo have been studied intensively and can be identified via specific exosomal-markers (e.g., tetraspanins (CD63, CD81), tumor susceptibility gene 101 (TSG101), or the 70-kDa heat shock protein 70 (HSP70) [6], MVs are more heterogeneous in their composition, and, as such, a ubiquitous set of specific MV markers has yet to be clearly defined [3]. As an additional confounding factor, it has been shown that the proteomic profiles of EVs from the same source depend on their isolation method [4], which may be partly explained by the lack of standardization of isolation procedures and specific markers for differentiating between Exo and MV and, after that, into their subgroups. The lack of specific markers, overlapping sizes, and physical properties of different subtypes of EVs make the isolation of enriched EVs an extremely challenging task, requiring the combination of different methodological approaches and techniques [7,8], which are often beyond the reach and scope of most research laboratories, especially those working in clinical settings. The purpose of this study was to establish an easy-to-use, reproducible, and scalable workflow to efficiently analyze EVs in the context of liver disease. Here we show that EVs distribution in the sera of patients with nonalcoholic fatty liver disease (NAFLD) or autoimmune hepatitis (AIH) is significantly different from the EVs circulating in the sera of healthy donors. Significant differences were also found in the EVs distribution in an animal model for hyperbilirubinemia (Gunn rats, [9]) as well as in the sera of aging rats. Here we present an optimized enrichment method that is effective in separating lipid-encapsulated nanoparticles from soluble proteins, which remain in the supernatant. We demonstrated that the resulting EVs are enriched in exosomal markers and can be used directly for biomarker discovery, including quantification of microRNAs and cytokines, or used further in functional studies, such as monitoring cellular uptake of labeled EVs.

## 2. Results

### 2.1. Characterization of Extracellular Vesicles in Crude Sera

Emerging evidence suggests that the quantity and composition of secreted vesicles change in different pathological conditions [10,11]. Hence, the accurate characterization of EVs is essential to obtaining further knowledge on their biological relevance. Several techniques are available to characterize EVs [7,12], including flow cytometry (FC), nanoparticle tracking analysis (NTA), transmission electron microscopy (TEM), and dynamic light scattering (DLS). In order to study the impact of liver diseases on the distribution of secreted EVs, NTA was used in this study to monitor and compare EVs characteristics. NTA is a technique based on light scattering that enables the rapid sizing and enumeration of nanosized particles in solution [13]. Since its introduction, NTA has become a standard procedure to reproducibility measure EVs size and concentration in their native conditions [14,15]. Importantly, precision and reproducibility of NTA measurements were achieved by using a standardized procedure (Appendix A).

In order to evaluate the performance of the NTA in different experimental models, EVs size and concentration were measured in the sera of human patients with NAFLD, AIH, and healthy donors, in sera isolated from young and aged rats, and in the sera obtained from Gunn rats with homozygote *Ugt1a* mutation and control rats (Figure 1, Representative NTA images are shown in Appendix A).

Here we show that vesicles circulating in patients’ sera were significantly smaller (CTRL vs. AIH, *p* = 0.0128) and more abundant (CTRL vs. NAFLD, *p* = 0.0027; CTRL vs. AIH, *p* = 0.0155) compared with EVs present in the sera of the healthy donor group (Figure 1a and Appendix A). Notably, a similar reduction of EVs size and a trend toward increased EVs number were observed also in sera prepared from aging rats (young vs. old rats size, *p* =0.0378, Figure 1b). Moreover, a similar trend was observed in an animal model for hyperbilirubinemia [9], which also displayed smaller (CTRL vs. Gunn rats, size, *p* = 0.0053) and more abundant EVs (CTRL vs. Gunn rats, number, *p* < 0.0001, Figure 1c and Appendix A).

To determine whether the distribution (e.g., size and number) of EVs circulating in patients with NAFLD and AIH correlates with liver injury, EVs-parameters were compared with the available baseline of liver enzymes measured in the patients’ sera (Table 1), but no significant correlation was found for the AIH patients (Appendix A). Regarding the NAFLD patients, a positive correlation was found between EVs size and the levels of glutamyl transpeptidase (GTP; R^2^ = 0.3779, *p* = 0.039, Figure 1d upper panel) and glutamic-oxaloacetic transaminase (GOT; R^2^ = 0.161, *p* = 0.0795, Figure 1d lower panel). Of note, a negative correlation was observed between EVs number and GTP (R^2^ = 0.1398, *p* = 0.1044, Figure 1e upper panel) and GOT (R^2^ = 0.1667, *p* = 0.0739, Figure 1e lower panel).

### 2.2. Assessing the Effect of Bilirubin, Hemoglobin and Proteins Aggregates on NTA Measurements

Due to the principles through which NTA achieves the detection and quantification of particles in suspension, NTA-detected signals in complex biological fluids could be generated either by cell-derived nano-vesicles (i.e., EVs and lipoproteins) or by other structures, including aggregates of serum proteins. NTA may also detect the presence of autofluorescent proteins such as bilirubin (Br) or hemoglobin (Hb).

EVs are readily lysed by the addition of detergents such as Triton X-100 [16]. Hence, to assess whether protein aggregates are present and detectable in serum, sera were incubated with either Proteinase K (PTK) or Triton X-100 (TX100), a combination of both (PTK/TX100), or left untreated. Samples were incubated either at room temperature (CTRL and TX100) or at 48 °C (CTRL-48 °C, PTK, and TX100/PTK). The effect of PTK and TX100 treatments on proteins was assessed by Western blot (Figure 2a). Here we show that PTK-mediated digestion of proteins did not significantly affect the number nor the size of EVs as measured by NTA (Figure 2b). However, TX100 alone or in combination with PTK administration did result in the partial (TX100) or complete (TX100/PTK) removal of signal detected by NTA compared with both control sera and PTK digested sera.

In healthy individuals, Br, a product of heme catabolism, is present at concentrations below 2 mg/dL. However, in individuals affected by biliary obstruction or by Crigler-Najjar syndromes, Br can rise up to 20 mg/dL (or 340 μmol/L, [17]). One of the animal models for the Crigler-Najjar syndrome is the Gunn rat, which closely parallels the genetic lesion observed in humans [9]. Analysis of EVs in Gunn rats’ sera identified significant differences in both the number (*p* < 0.0001; Gunn; 4.17 × 10^12^ ± 1.76 × 10^11^ vs. CTRL; 2.83 × 10^11^ ± 2.33 × 10^10^) and the size (*p* < 0.005; Gunn; 126 ± 1.18 nm vs. CTRL; 135.3 ± 1.19 nm) of EVs (Figure 1c). To assess whether the observed differences were disease-driven or due to higher-than-normal levels of Br, solutions containing different amounts of Br were analyzed (Figure 2c). Importantly, NTA did not detect any signal even when solutions containing 10 mg/mL Br (i.e., 1.0 g/dL, 10 mg/mL BR solution is shown in Figure 2d), which is over 400 times the amount of bilirubin measured in Gunn rat sera [4.1 ± 1.25 mg/dL (n = 4); CTRL rats: 0.13 ± 0.009 mg/dL (n = 9; Figure 2c)].

As a consequence of red blood cell rapture, hemolysis may occur during blood collection, and cytosolic content, including Hb, may be released in the serum. To evaluate whether hemolysis might contribute to the NTA signal, hemoglobin was dissolved to simulate different levels of hemolysis, from 0.5% to 50% (Figure 2d). Here we show that Hb becomes only detectable by NTA at a concentration of 10 mg/mL (equivalent to ~5% hemolysis, 10 mg/mL Hb solution is shown in Figure 2d) or higher. These data suggest that Hb is unlikely to contribute to NTA measurements in those samples where hemolysis is not detectable by visual inspection. Based on these findings, we concluded that protein aggregate, or Br, is unlikely to contribute to the quantification of EVs by NTA in non-hemolyzed crude sera. Nevertheless, it is recommended for hemolyzed specimens to be discharged.

### 2.3. Separation of EVs from Serum-Protein by Using an Optimized PEG Approach

PEG has been used to purify viral particles for more than 60 years [18]. It was previously shown that PEG can also be used to enrich EVs from biological fluids [19], but when we employed the method as described by Rider et al., a significant amount of serum-protein contaminant (e.g., albumin) was detected in the EVs-enriched pellets (Appendix A). In order to reduce the quantity of contaminating proteins, a PEG titration curve was carried out. Here we show that a 6:1 (*v*/*v*) dilution of serum (or plasma) with PEG Precipitation Solution (PPS) stock resulted in the removal of >99% of soluble proteins from the EVs-enriched pellets. Notably, PPS performance is comparable to commercial EVs isolation reagents such as TEI (Figure 3a and Appendix A). Importantly, when the enriched EVs were subjected to a second cycle of PPS isolation, no albumin was detected in the resulting pellets. The purity of the EVs preparation was evaluated by Western blot analysis of the exosomal markers TSG101 (Figure 3b), CD63, and HSP70 (Appendix A) [6], which were exclusively detectable in the EVs-enriched pellets, whereas the serum protein albumin was predominantly detected in the EVs-depleted supernatant (Figure 3c and Appendix A). Overall, these data support the conclusion that the presented approach efficiently separates EVs from soluble serum proteins.

### 2.4. Evaluation of Extracellular Vesicles Integrity and Functionality

To assess the integrity of PPS-isolated EVs, measurement of the cellular uptake of fluorescently labeled EVs was performed. For this purpose, EVs were fluorescently labeled with either Syto RNASelect (FITC-label) or Cell-Mask marker (PE-label). EVs labeling was assessed with a BD FACSAria III flow cytometer (FC). Because the FC in use in our laboratory lacks dedicated software to quantify the size and number of nano-sized particles, the following empirical approach was used (previously described in [20]). For this purpose, fluorescently labeled particles of defined sizes (100 nm, 200 nm, and 600 nm) were loaded on the FC, enabling the establishment of a 200 nm gate generated with the 200 nm FITC-labeled beads (Figure 4a,b). Following this, sera containing labeled EVs were loaded on the FC, and the 200 nm gate was used to visualize both Syto RNASelect (Figure 4c) and Cell-Mask (Appendix A) labeled EVs.

Furthermore, sera containing fluorescently labeled RNAs were either digested with PTK or mock digested before PEG-mediated precipitation. EVs-enriched pellets were then dissolved in serum-free culture medium and supplemented with rat PCs in culture. The uptake of EVs carrying fluorescent RNAs was visualized by microscopy (Figure 4d), and fluorometric analysis was carried out to quantify the uptake of fluorescently labeled RNAs (Figure 4e). The observed increase in fluorescence measured in PCs incubated with EVs carrying SytoRNA Select Labeled RNAs supports the conclusion that the exogenous RNA was transferred from the EVs to the cytosol of these cells. Importantly, because cells incubated with EVs digested with PTK showed only a modest increase in fluorescence, we can conclude that interaction between the surface markers of EVs and receptors on the membrane of PCs is required. Overall, these data support the conclusion that isolation with PEG preserves the integrity of EVs and that PEG-enriched EVs are suitable for functional study.

### 2.5. Analysis of miRNA Expression in EVs Isolated from Patients with Liver Diseases and Aged Animals

In the last decade, cell-free miRNAs in body fluids have emerged as a new class of biomarkers for various diseases, including liver diseases [21,22]. Cell-free miRNAs are both associated with EVs and RNA-binding proteins (RBPs). Notably, electropherograms of RNAs isolated from EV-enriched pellets and EV-depleted supernatants display distinctly different RNA profiles (Figure 5a), which underscores the importance of separating the two sets of RNAs as they may possess different physiological functions.

To assess whether PPS-isolated EVs are suitable material for the identification of potential disease-associated biomarkers, a group of 11 miRNAs (Appendix A) was selected through the mining of the publicly available study GSE113740 [22] that contains the profiling of cell-free miRNAs circulating in the sera of patients with chronic hepatitis, liver cirrhosis, and HCC (Appendix A). Of the selected miRNAs, a total of five miRNAs (i.e., miR-15b, -16, -26a, -122, and -148a) were found to be significantly up-regulated in EVs isolated from AIH and NAFLD patients compared with the CTRL groups (Figure 5b, top panel, and Figure 5c). The expression of another three miRNAs (i.e., miR-142-3p, -10a, and -223, Figure 5b, middle panel, and Figure 5c) was found to be significantly upregulated in EVs isolated from AIH patients compared with the healthy donors, whereas the expression of another three miRNAs (i.e., miR150, -15a, and -21, Figure 5b, lower panel, and Figure 5c) was found significantly upregulated in AIH vs. NAFLD, AIH vs. CTRL and NAFLD vs. CTRL, respectively (see also Appendix A). Further studies in larger cohorts will be needed to effectively validate if this panel of miRNAs represents a liver-disease-based biomarker signature with diagnostic or prognostic value.

We have previously shown that hepatic stellate cells (HSCs) isolated from the livers of aged rats exhibited a senescence-associated secretory phenotype and lowered expression of matrix proteins and growth factors compared with HSCs isolated from the livers of young animals [23]. NTA analysis of EVs circulating in the blood of these animals revealed that the EVs distribution between old and young animals is also significantly different. (Figure 1). Now we show that the levels of several miRNAs that were found to be increased in patients with NAFLD and AIH (Figure 5b) were also found to be significantly different between EVs isolated from the sera of older rats and younger animals (i.e., miR-122, -155, and -142-3p UP and miR-150 and -18a DOWN, Figure 5c). It would be interesting to extend this analysis to assess whether these differences in EVs distribution and miRNAs levels may actually reflect a signature of healthy aging or instead emphasize conditions of cellular senescence and inflammation.

### 2.6. Luminex-Based Measurement of Cytokines in EV-Enriched Pellets and EVs-Depleted Supernatants

Cytokines are important modulators of immune function and inflammatory responses that function as soluble factors mediating cell-cell communications in multicellular organisms. It has been reported that cytokine secretion occurs both classically and through encapsulation in EVs [24,25]. One of the general aims of our work is the identification of early, non-invasive diagnostic biomarkers of cholangiocarcinoma (CCA). CCA has recently shown increasing mortality rates [26]. Therefore, there is an urgent need for biomarkers for early diagnosis of CCA. Interestingly, cytokines are among the emerging diagnostic and prognostic biomarkers in CCA [27], and IL-6, IL-8, and granulocyte-macrophage colony-stimulating factor (GM-CSF) have been found to be increased in CCA patients [28]. In order to benchmark the PEG-mediated isolation method with respect to the possibility of analyzing cytokines in EVs, the following pilot study was conducted. EVs were isolated from sera of 16 patients with CCA and from 5 healthy donors, and a Luminex 8plex was used to measure cytokines in whole sera (WS) and lysed EVs (LyEVs). Notably, differences were observed in both the type and range of cytokines detected (Figure 6a,b and Appendix A). While for certain cytokines the analysis of both WS and LyEVs material generated comparable results (i.e., IL-8 and IL-6), for others the analysis of LyEVs material appears to provide more information. For instance, the levels of IL-2 were identified as significantly increased in the CCA group only in the samples prepared by the LyEVs approach. Moreover, GM-CSF and INFγ were exclusively detected in the samples prepared according to the LyEVs approach, although no significant differences were found between the CCA and the control groups. Overall, our data indicate that PEG-mediated isolation of EVs is compatible with cytokine analysis by Luminex. Furthermore, the data presented here suggest that (for the cytokines included in the 8plex) the LyEVs approach might be a better source of material for the discovery of diagnostic biomarkers in CCA.

## 3. Discussion

Compelling evidence indicates that EVs play relevant roles in cell-cell communication and may contribute to the pathogenesis of human diseases. Numerous studies show that the isolation of EVs by existing methodologies may result in considerable carry-over of contaminants in the form of serum proteins and lipoproteins [29,30]. Therefore, enrichment of “cleaner” EVs requires the combination of two or more isolation methods, including several control steps. While this strategy is a must when characterizing different subpopulations of EVs, it renders the task of characterizing EVs (e.g., measuring their size and number) a daunting task. In order to assess if EVs can be directly measured in crude serum by NTA, an extensive number of control experiments were performed, and based on these experiments, we conclude that serum-proteins, Br or Hb, do not contribute to NTA measurements. These conclusions are corroborated by the observation that NTA detects no signals in EVs-depleted sera, which contains ≥99% of serum proteins. We show that NTA analysis of EVs in the sera of patients with liver disease identified significant differences in both EVs size and number compared with the control group (Figure 1a). Interestingly, similar changes in EVs distribution were also observed in aged compared with younger animals (Figure 1b) and in an animal model for bilirubinemia (Figure 1c). These findings are in agreement with the knowledge that inflammation might influence the machinery responsible for EVs biogenesis or secretion [10]. Hirsova et al. showed that primary hepatocytes incubated with lysophosphatidylcholine (LPC, lipotoxic compound) secrete a significantly higher number of EVs [31], while Kakazu et al. demonstrated that stimulation of immortalized mouse hepatocytes with palmitic acid increase EVs secretion compared with the vehicle alone [32]. In a previous study, we showed that administration of inflammatory cytokines to primary hepatic stellate cells affects both the amount and size of secreted EVs [33]. To the best of our knowledge, the current work is the first to report a quantitative analysis of the distribution of EVs in patients with chronic liver disease, and it supports the conclusion that analysis of EVs in crude serum might provide prognostic or diagnostic information. In this regard, we have recently shown that EVs sizes represent a novel prognostic marker in patients receiving transarterial chemoembolization (TACE) for primary and secondary hepatic malignancies [34].

In this study, we also established a cost-effective method to separate EVs from serum proteins (Figure 3 and Appendix A) and benchmarked it against the identification of known serum miRNA biomarkers for liver diseases [21,22]. In agreement with published literature (Appendix A, [22]), we show that the levels of several EVs-associated miRNAs were significantly increased in patients with NAFLD and AIH but not in the control group (Figure 5b). Specifically, miR-142-3p, miR-10a, miR-150, and miR-223 were significantly higher in EVs isolated from patients with AIH compared with healthy donors. Furthermore, the levels of miR-15a and miR-21 were found to be significantly increased in EVs circulating in AIH compared with NAFLD patients. Although a major limitation of our study is the small number of patients involved, which we recognize could potentially lead to biased results, we would like to emphasize that our intention in promoting this work is not to present it as conclusive evidence but rather to demonstrate the effectiveness of the methodology employed and to illustrate the potential benefits that the application of this method could have in the clinical setting. Future studies in large multi-center cohorts will be necessary to determine whether the combination of these miRNAs may be used as a reliable diagnostic or prognostic marker in NAFLD, AIH, and potentially other liver diseases.

Interestingly, striking similarities in EVs distribution and miRNAs levels were also observed when comparing young and aged rats (Figure 1b and Figure 5c). These data emphasize the importance of matching donor age, as some of the observed differences may actually indicate a difference in age rather than indicating ongoing disease. As previously shown by others [35,36], PEG-enriched EVs were also found to be enriched in small RNAs (Figure 5a). However, striking differences were observed in the distribution of RNAs extracted from enriched EVs and EV-depleted sera. This observation underscores the importance of separating EVs from soluble proteins and supports the conclusion that PEG-enriched EVs can serve as an entry point for further downstream procedures aimed at enriching specific EV subtypes.

Besides miRNAs, EVs transport other types of biologically active molecules. To this end, it was shown that cytokines are present in EVs [24,25] and that some of the EVs-encapsulated cytokines are not detectable by standard assays [25]. As cytokines are emerging candidate biomarkers in human diseases, including cholangiocarcinoma [26,27,28], we investigated whether PEG-enriched EVs could be a suitable source for analyzing EVs-encapsulated cytokines. Here we show that not only is our method compatible with cytokine detection and measurement by Luminex (Figure 6), but also that cytokines contained in EVs display a better dynamic range compared with cytokines circulating in whole serum. Furthermore, Luminex was able to detect INFγ and GM-CSF cytokines only in LyEVs samples. Although cytokines have already been measured in cancer-derived EVs [37], to our knowledge, this is the first study in which cytokines were profiled in serum-derived EVs isolated from CCA patients.

In conclusion, we presented an optimized and easy-to-use workflow that enables qualitative and quantitative characterization of EVs in complex biological fluids and their enrichment for subsequent analysis, including biomarker and functional characterization.

## 4. Materials & Methods

### 4.1. Quantification of EVs by Using Nanoparticle Tracking Analyzer and Flow Cytometry

Analysis of EVs was performed with the ZetaView multi-parameter Particle Tracking Analyzer (ParticleMetrix, Wildmoos Germany). Samples were diluted in PBS to achieve a particle count in the range of 1–9 × 10^7^ p/mL (or 250 to 300 particles per visual field, PVF). Using the script control function, five 30-s videos for each sample were recorded, incorporating a sample advance and a 5-s delay between each recording. For analysis of bilirubin (Br) and hemoglobin (Hb), samples were diluted in PBS according to the same dilution factor as sera from control animals (i.e., 1:7000). Bilirubin (Br) stock was prepared in chloroform to a final concentration of 10 mg/mL. For estimating the amount of hemolysis, it was considered that red blood cells (RBCs) circulating in a healthy human adult contain 150–200 mg of hemoglobin per mL of blood (or 15–20 g/dL). Human hemoglobin (Sigma-Aldrich, Taufkirchen, Germany) was dissolved in PBS at 10 mg/mL or 1 mg/mL. For achieving 100 mg/mL, Hb stock was diluted 1:700 (instead of 1:7000) in PBS before NTA measurement. Furthermore, to mimic hemolysis, 0.5% (1 mg/mL), 5% (10 mg/mL), or 50% (100 mg/mL) Hb dilutions were prepared in PBS.

In order to visualize EVs by Flow Cytometer (FC), sera samples and conditioned tissue culture medium from rat hepatic stellate cells (HSCs) or primary rat hepatocytes (PCs) were stained with phycoerythrin (PE)-labeled Cell Mask (Thermo Fisher Scientific, Meerbusch, Germany) or stained with Syto RNASelect (Thermo Fisher Scientific, Meerbusch, Germany), measured in the FITC channel. Using a FACSAria III Cell Sorter (BD Biosciences, Heidelberg, Germany) equipped with four air-cooled lasers at 375-, 488-, 561-, and 633-nm wavelengths, a 1.0 neutral density filter in front of the forward scatter detector was used to decrease the forward scatter (FSC) signal. To avoid exclusion of the smallest events, an absolute minimum threshold of 200 was set at the side scatter (SSC)-A parameter (instead of FSC-A). To locate the size of 200 nm EVs, a gate was established using size-calibrated and FITC-labeled beads with sizes of 200 nm (TetraSpeck, Thermo Fisher Scientific, Meerbusch, Germany, T7280). Samples were evaluated with Flowjo v 10 (FlowjoLLC, BD Biosciences, Heidelberg, Germany).

### 4.2. Isolation of extracellular vesicles with PEG and TEI, and labeling

Before precipitation, serum samples were precleared by centrifugation at 10,000 relative centrifugal force (rcf) for 30 min at 4 °C. Supernatants were transferred to a new tube, and sera were either left untreated (CTRL) or diluted with either the recommended amount of TEI [Total Exosome Isolation Reagent from serum (Invitrogen, Waltham, MA, USA, 4478360)] or different amounts of PEG precipitation solution (PPS). In the initial testing, different amounts of stock solution (24% PEG-8000, 1.5 M NaCl) were added to the sera to obtain the final amounts of PEG 4%/250 mM NaCl (1:6 dilution), PEG 8%/500 mM NaCl (1:3 dilution), and PEG 12%/750 mM NaCl (1:2 dilution). The samples were incubated for 30 min at 4 °C and then centrifuged (10,000 rcf for 30 min at 4 °C). The vesicle-free supernatants were transferred to fresh tubes for NTA or Western blot analysis, while the vesicle-enriched pellets were rinsed twice with 1 mL of cold PBS and finally dissolved in PBS for downstream procedures. A ready-to-use protocol for the preparation and use of PEG solution has been made available as a Appendix A. For labeling of EVs, 1 μL of either Cell Mask orange plasma membrane stain (Thermo Fisher Scientific, Meerbusch, Germany, C10045) or Syto RNASelect (Thermo Fisher Scientific, Meerbusch, Germany, S32703) was added to 500 μL of serum and incubated for 15 min at room temperature (RT) before treatment with Proteinase K (Carl Roth, Karlsruhe, Germany, 7528.1) or Triton X-100 or the addition of precipitation reagents. For digestion/lysis EVs-enriched pellets were dissolved in PBS and either incubated with 200 μg/mL of proteinase K for 1 h at 48 °C or with 0.3% Triton X-100 for 15 min at RT. Labeled/digested EVs were precipitated with PEG as described above.

### 4.3. Western Blot Analyses and Antibodies

Protein concentrations were measured using the Qubit Protein Assay Kit (Thermo Fisher Scientific, Meerbusch, Germany, Q33211) according to the manufacturer’s instructions. Western blot analyses were carried out using 20 μg of total proteins. Serum proteins were first precipitated by the addition of 10% (*w*/*v*) trichloroacetic acid (TCA; Sigma-Aldrich, Taufkirchen, Germany, 49010) and incubated at −20 °C for up to 30 min. Denatured proteins were pelleted by centrifugal force (10 min, 4 °C at 10,000 rcf) and washed shortly with cold acetone. Following centrifugation (10 min, 4 °C at 10,000 rcf), acetone was removed, and the protein pellet was shortly allowed to dry. The proteins were then redissolved in loading buffer (250 mM Tris pH 6.8, 10% (*v*/*v*) glycerol, 2% SDS, 0.01% (*w*/*v*) bromophenol blue, 50 mM DTT). Protein lysates were loaded together with 10 μL of Protein Ladder (BioRad, Dusseldorf, Germany, 161-0373) on 10% or 12% SDS polyacrylamide gels and transferred to nitrocellulose membranes using semidry blotting systems according to standard protocols. Membranes were blocked with 5% milk powder (Carl Roth, Karlsruhe, Germany, T145.3) in Tris-buffered saline with Tween20 (TBST) for 1 h at RT or overnight at 4 °C, followed by incubation with a horseradish peroxidase (HRP)-coupled antibody against rat albumin (Bethyl Laboratories, Montgomery, TX, USA, A110-134P) for 2 h at room temperature. Chemiluminescence was detected with ECL Western blotting Substrate (Promega, Walldorf, Germany, W1001) using the ChemiDoc MP Imaging System (BioRad, Dusseldorf, Germany). Signal intensities of Western blot protein bands were analyzed using ImageJ 1.51 software (National Institute of Health, Bethesda, MD, USA). Antibodies used in protein analysis by Western blot: anti-Albumin antibody (1:2000 dilution, Abcam, Cambridge, UK, ab207327), anti-TSG101 antibody (1:5000 dilution, Abcam, Cambridge, UK, ab125011), anti-CD63 antibody (1:200 dilution, BioRad, Dusseldorf, Germany, MCA4754GA), and anti-Hsp70 antibody (1:500 dilution, Cell Signaling, Danvers, MA, USA, 4872T).

### 4.4. RNA Isolation, Visualization and miRNA Analysis by miQPCR

Previous reports indicate that the concentration of cell-free RNA circulating in blood is low, in the range between 1 and 150 ng/mL [38,39], suggesting that the amount of RNA that can be recovered from 100–500 μL of serum might be below the detection limit of standard quantification methods. Indeed, we were unable to quantify RNAs by using either photometric (NanoDrop, Nanodrop Technologies, Wilmington, DE, USA) or fluorimetric (Qubit, Thermo Fisher Scientific, Meerbusch, Germany) approaches. In this work, EVs were either enriched from 100 or 500 μL of serum. EVs-enriched pellets were dissolved in 100 μL of PBS, and RNAs were isolated using the miRNeasy Mini Kit (Qiagen, Hilden, Germany, 217004). In order to generate an electropherogram (2100 Bioanalyzer, Agilent Technologies, Santa Clara, CA, USA), EVs were enriched from 500 μL of sera, and RNAs were isolated from both EVs-enriched pellets and EVs-depleted sera and eluted in 100 μL of elution buffer. Following this, 1 μL of the resulting eluted material was loaded and run on a PicoChip according to the supplied protocol. For the analysis of miRNA expression, EVs were enriched from 100 μL of sera, and RNA from EVs-enriched pellets was recovered in 40 μL of elution buffer. cDNA synthesis was carried out by using equal volumes of eluted material as described in [40]. qPCR reactions were carried out on a StepOnePlus Real-Time PCR system (Thermo Fisher Scientific, Meerbusch, Germany), whereas amplicons were detected using SYBR Green I (Promega, Walldorf, Germany, A6002). Data analysis was carried out by using qBase [41], and GeNorm was used to identify appropriate reference genes. Gene expression was normalized to Let-7a, Let-7c, and Let-7e by using the ΔΔCt method [42]. The sequences of the miQPCR primers designed to amplify miRNAs included in this study are listed in Appendix A.

### 4.5. Preparation of Primary Cells from Rat Liver and Liver Samples from Gunn and Aging Rats

Primary hepatocytes were isolated from the livers of 2-month-old male Wistar rats (150–200 g) essentially as described in ref. [40]. In brief, hepatocytes were isolated after serial perfusion of rat liver by Hanks’ balanced salt solution (HBSS, Sigma-Aldrich, Taufkirchen, Germany, H6648) and collagenase CLS type II solution (50 mg/150 mL, Biochrom, Terre Haute, IN, USA, C2-22). Following digestion and centrifugation, pelleted hepatocytes were resuspended in culture medium (Williams’ E medium, Millipore, Darmstadt, Germany, F1115) supplemented with 10% (*v*/*v*) fetal calf serum (FCS superior, Millipore, S0615), 2 mM *L*-glutamine (Gibco, Billings, MT, USA, 25030), 0.6 μg/mL insulin (Sigma-Aldrich, Taufkirchen, Germany, I05116), 100 nM dexamethasone in DMSO (Sigma-Aldrich, Taufkirchen, Germany, D8893), and 1% (*v*/*v*) penicillin-streptomycin-amphotericin B solution (Gibco, Billings, MT, USA, 15240-062). Hepatocytes were seeded on polystyrene tissue culture dishes of appropriate size, pre-coated with collagen type I (Sigma-Aldrich, Taufkirchen, Germany, C3867, 6–10 mg/cm^2^).

The homozygous mutant Wistar rat strain (Gunn-UGT1A1; [43]) lacking the uridine diphosphate glucuronosyltransferase-1A1 (UGT1A1) enzyme activity and exhibiting elevated bilirubin serum levels was used in this study. The total bilirubin concentration of the Gunn rat serum was determined by the central hospital laboratory of Heinrich Heine University. Gunn rats were obtained from the Rat Resource & Research Centre (RRRC, Columbia, MO, USA) and maintained at the animal facility of the Heinrich Heine University (Düsseldorf, Germany). Normal Wistar rats were obtained from Janvier Labs (Le Genest-Saint-Isle, France) for liver aging studies. Young (2-month-old) and aged rats (22-month-old) from the same breeding colony with matching housing conditions were used to obtain blood serum samples after cannulation of the portal vein.

### 4.6. Visualization and Quantification of EVs Uptake

Primary hepatocytes isolated from rat livers were plated in 24-well plates coated with collagen type I in William’s E (WE) medium supplemented with 10% FCS and allowed to attach and recover for 24 h. Thereafter, cells were incubated in serum-free WE medium containing ~2.0 × 10^11^ of fluorescently labeled EVs with/without Proteinase K (PTK) digestion or a negative control (i.e., mock labeling) and cultured for 24 h at 37 °C and 5% CO_2_ in a humidified atmosphere. Furthermore, the culture medium was exchanged, and the cells were incubated for another 24 h. Images of live cells were taken with an Olympus IX50 fluorescence microscope equipped with a DP71 digital camera (Olympus, Hamburg, Germany) using the excitation filter set 470/22 nm and the emission filter set 510/42 nm.

For the quantification of EVs uptake, background fluorescence was assessed by measuring fluorescence before the addition of labeled EVs (Time 0). Following the addition of EVs, a measurement was taken every 90 min for 5 times. Fluorimetric quantification was carried out by a microplate fluorimeter (Fluoroskan Ascent FL, Thermo Fisher Scientific, Meerbusch, Germany) after excitation at 485 nm. The emitted fluorescence light was measured at 538 nm. The data were analyzed using Microsoft Excel 2016 and GraphPad Prism (version 9.4.1).

### 4.7. Patient’s Material and Cytokine Quantification by Luminex

NAFLD and AIH samples included in this study were from patients visiting the hepatitis outpatient clinic of the University Hospital of Düsseldorf (i.e., not hospitalized). The hepatitis outpatient clinic is a trans-regional center for patients with chronic viral diseases of the liver. Only sera from patients negative for viral hepatitis were included in this study. Healthy volunteers from our department donated the blood for the control group. The analysis of liver parameters was carried out by the central hospital laboratory of Heinrich-Heine University.

The CCA cohort consists of n = 16 patients with intrahepatic cholangiocarcinoma (CCA) who were admitted to the Department of Visceral and Transplantation Surgery at University Hospital RWTH Aachen for tumor resection and were recruited between 2011 and 2017 (see Appendix A for patient characteristics). Whole blood samples were taken before surgery, centrifuged for 10 min at 2000 g, and serum samples were then stored at −80 °C until use. Billiary tract cancer (BTC) was confirmed histologically in the resected tumor sample. For the preparation of serum from healthy controls, whole blood samples were taken at RT and collected into Sarstedt collection tubes (serum: S-Monovette 7.5 mL Z/plasma: S-Monovette 2.7 mL K2 EDTA, both Sarstedt, Nümbrecht, Germany) and centrifuged for 10 min at 2000 rcf. The supernatant (serum or plasma) was carefully transferred into a fresh tube and stored at −80 °C until use.

EVs isolation for Luminex quantification of cytokines in sera from the CCA cohort was carried out using PEG 4%/250 mM NaCl (1:6 dilution) as described above. To ensure the complete removal of serum proteins, the procedure was repeated three times. Following the third precipitation, EVs were diluted in 25 μL of lysis buffer (0.3% Triton X-100 in PBS) and incubated for 15 min at RT. Samples were analyzed by multiplex immunoassay according to the manufacturer’s instructions using a Bio-Plex 200 system and Bio-Plex Manager 6.0 software on a Bio-Plex Pro Human Chemokine Panel (BioRad, Dusseldorf, Germany, 171AK99MR2), including the cytokines: IL-2, IL-4, IL-6, IL-8, IL-10, GM-CSF, INFγ, and TNFα. A serum sample volume of 50 μL, or an amount of enriched EVs equivalent to 50 μL of serum, was used.

### 4.8. Statistical Analysis and Imaging Software

Statistical analyses were carried out using GraphPad Prism (version 9.4.1). To evaluate whether samples were normally distributed, the D’Agostino and Pearson normality tests were carried out. When the sample distribution passed the normality test, then parametric tests were carried out (i.e., one-way analysis of variance/ANOVA for three or more samples and a two-tailed *T*-test for two samples). When the samples did not pass the normality test, non-parametric tests were applied (i.e., the Kruskal-Wallis test for three or more samples and the Mann-Whitney test for two samples). The data were considered significant at a *p* value ≤ 0.05. Images were prepared using Affinity Designer (version 1.10.4.1198).

## Figures and Tables

**Figure 1 ijms-24-09631-f001:**
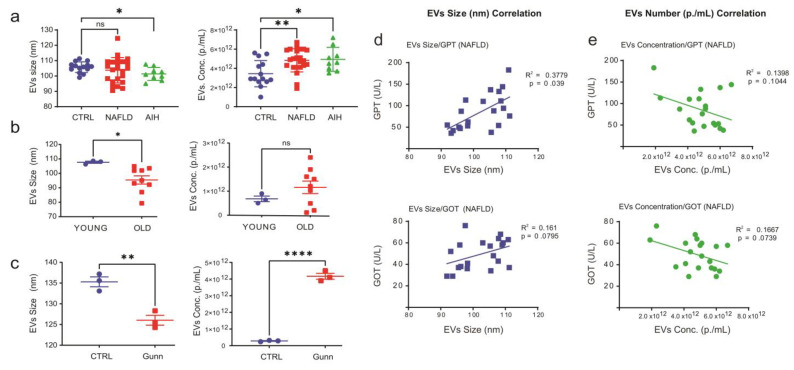
NTA quantification of EVs in sera of liver-diseased patients. Sera from patients with either NAFLD (n = 24), or AIH (n = 9) or healthy donors (n = 14) were analyzed by NTA. (**a**) Average particle size as measured by NTA showed a reduced EV size in AIH patients compared with healthy individuals, while quantification of EVs in patients’ sera indicated an increased number of EVs in NAFLD and AIH patients compared with control, respectively. (**b**,**c**) Analysis of EVs size distribution and number in the sera of (**b**) younger (2-months-old) and older (22-months-old) rats (n = 3 and n = 9, respectively) and in the sera of (**c**) Gunn and control rats (n = 3). Correlations of serum GOT and GPT, with particle average sizes (**d**) and EV number (**e**) in NAFLD patients. Statistical analyses were carried out in GraphPad using one-way ANOVA or two tailed unpaired t-tests for comparison of two groups. ns = Not significant; *p* ≤ 0.05, *; *p* ≤ 0.01, **; *p* ≤ 0.0001, ****.

**Figure 2 ijms-24-09631-f002:**
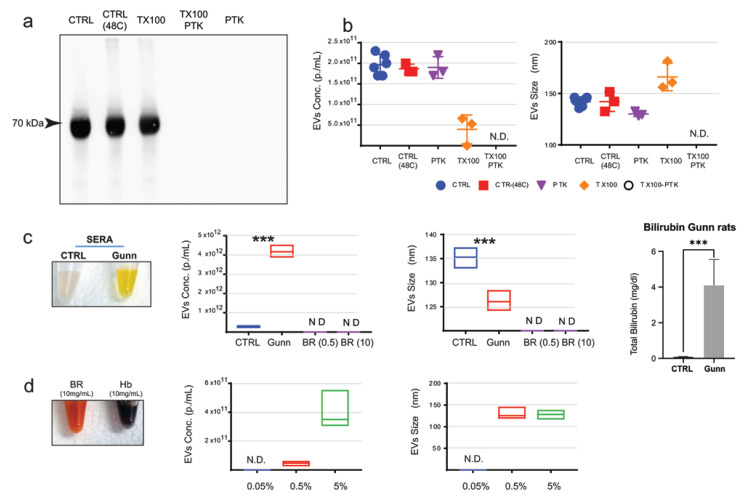
Evaluation of the effect of bilirubin, hemoglobin, and protein aggregates on NTA measurements: (**a**) Western blot for albumin in rat sera treated with either 500 μg/mL Proteinase K (PTK), 1% Triton X-100 (TX100), or the combination of the two (PTK/TX100) and incubated at 48 °C overnight. Control samples were left untreated (CTRL) or incubated at 48 °C overnight [CTRL(48C)]. The figure shows protein bands with 70 kDa size (integration time 50 s). (**b**) NTA analysis of the PTK, TX100, PTK/TX100 treated or control sera (n = 6). (**c**) EVs circulating in sera from Gunn and control (CTRL) rats were analyzed by NTA. EVs concentration (**left** panel) and size (**right** panel) in the sera of CTRL and Gunn rats, showing a significant increase of EVs concentration and reduction in EVs size in Gunn compared with CTRL rats (n = 3). To evaluate the potential effects of Bilirubin (Br), NTA measurement in crude rat serum (or plasma), Br was diluted at 0.5 mg/mL [BR(0.5),] and 10 mg/mL [BR(10)] in PBS (see material and method section) and measured by NTA. No signal was detected by NTA in Br solutions. (**d**) To evaluate the potential effects of hemolysis of NTA, hemoglobin (Hb) was diluted in PBS to simulate 0.5% and 5% hemolysis (see material and method), and measured by NTA. Data are shown as average ± standard deviation (n = 5), statistical analyses were carried out in GraphPad using one-way. N.D. = Not detectable; *p* ≤ 0.001, ***.

**Figure 3 ijms-24-09631-f003:**
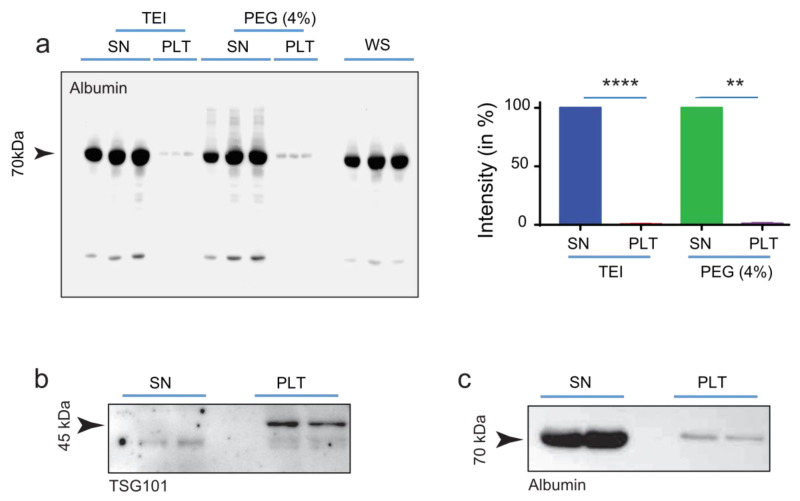
PEG-isolated EVs are enriched in exosomal markers. (**a**) Quantification of serum protein carry-over. TEI precipitation resulted in 0.61% (±0.161, n = 3), whereas PEG (4%) resulted in 1.04% (±0.466, n = 3) carry-over of serum albumin in the pellets (PEL) (n = 3). (**Top** panel) after 1 cycle of precipitation about 1% albumin was still detectable in the EVs-enriched pellet. (**b**) The exosomal markers TSG101 was exclusively detected in the EVs enriched pellets (PLT), on the other hand, (**c**) albumin was mainly detected in the EVs-enriched pellet (SN). Statistical analyses were carried out in GraphPad using two tailed unpaired t-tests for comparison of two groups. *p* ≤ 0.01, **; *p* ≤ 0.0001, ****.

**Figure 4 ijms-24-09631-f004:**
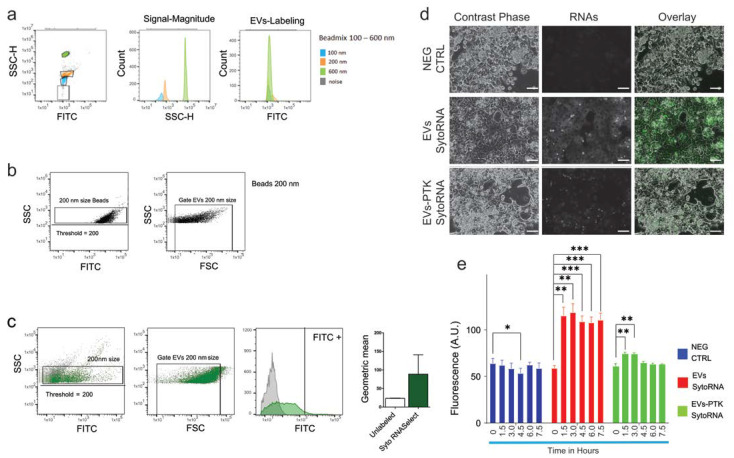
Evaluation of extracellular vesicles integrity and functionality: (**a**) Fluorescently labeled particles of defined size (100 nm, 200 nm and 600 nm), were loaded on the BD FACSAria III flow cytometer (FC), and (**b**) EVs of 200 nm in size were used to establish of a 200 nm gate. (**c**) Sera containing Syto RNASelect (FITC-label) labeled EVs were loaded on the FC and the 200 nm gate was used to visualize and quantify the number and fluorescence of Syto RNASelect labeled EVs. (**d**) Uptake of EVs by hepatocytes is shown by the increase in green fluorescence (middle panel, Syto RNASelect-labeled RNA), the increase in fluorescence is absent from cells incubated with EVs that were digested with PTK. Images acquired by fluorescence microscopy after 24 h incubation. Scale bars 100 μm (**e**) Quantification of relative fluorescence measured from primary rat hepatocytes following the uptake of EVs containing fluorescently labeled RNAs with or without PTK digestion. Fluorescence was measured before EVs addition (time 0) and every 90 min for five times. Data are represent as average fluorescence ± SD (n = 8), only significant differences are shown. Statistical analyses were carried out in GraphPad using one-way ANOVA. *p* ≤ 0.05, *; *p* ≤ 0.01, **; *p* ≤ 0.001, ***.

**Figure 5 ijms-24-09631-f005:**
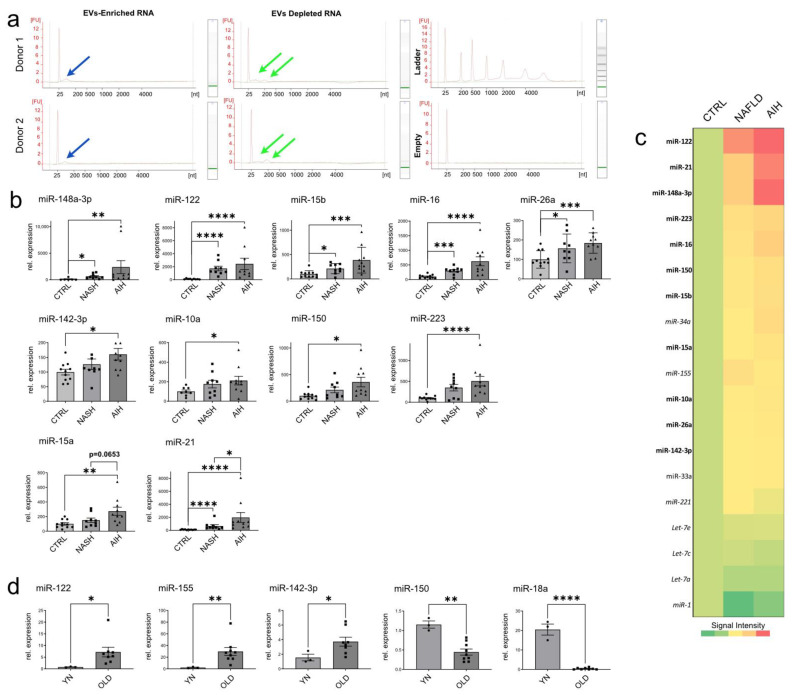
Analysis of miRNA expression in EVs isolated from patients with liver diseases and aged animals: (**a**) Electropherograms of RNAs isolated from EV-enriched pellets and EV-depleted supernatants display distinctly different RNA profiles. The blue (EVs enriched RNAs) and green (Evs depleted RNAs) arrows show the location of the RNA peaks on the electropherograms. (**b**) miQPCR was used to quantify the levels of miRNAs isolated from EVs-enriched pellets from liver-diseased patients. EVs were isolated from the sera of NAFLD patients (n = 11), AIH patients (n = 7) and healthy donors (CTRL; n = 11). (**b upper panel**) The expression of seven miRNAs was found significantly increased in both AIH and NAFLD vs. CTRL. (**b middle panel**) miR-142-3p, -10a and -223 expression was found to be significantly elevated in AIH vs. CTRL. (**b lower panel**) miR-150, -15a and -21 expression was found to be significantly elevated in AIH vs. NAFLD. (**c**) Heat map representation of the data shown in panel b. (**d**) Analysis of selected miRNAs in EVs isolated from the sera of young (2-months-old, n = 3) and old (22-months-old n = 9) rats. Data are represent as average ± SD. Statistical analyses were carried out in GraphPad using one-way ANOVA and two tailed unpaired *t*-tests for comparison of two groups. *p* ≤ 0.05, *; *p* ≤ 0.01, **; *p* ≤ 0.001, ***; *p* ≤ 0.0001, ****.

**Figure 6 ijms-24-09631-f006:**
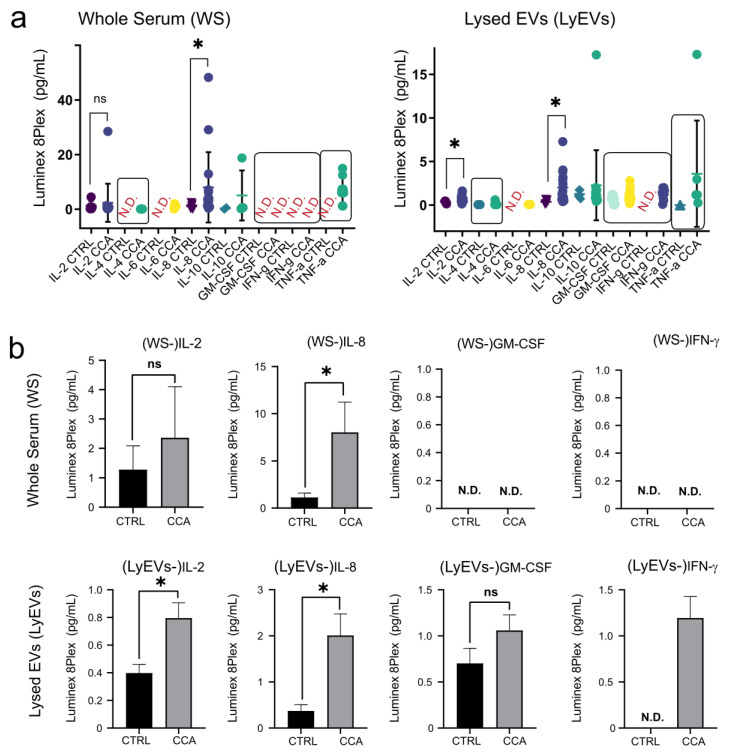
Luminex-based measurement of cytokines in EV-enriched pellets and EVs-depleted supernatants: (**a**) Quantification of cytokines in sera from the CCA cohort and healthy controls (CTRL) by 8plex Luminex in whole sera (WS, **left** panel) and in lysed EVs (LEVs, **right** panel). (**b**) Side by side representation of the levels of the selected cytokines in WS (**top** panel) and LEVs (**lower** panel). Statistical analyses were carried out in GraphPad using two tailed unpaired t-tests for comparison of two groups. ns = Not significant; N.D.= Not Detectable; *p* ≤ 0.05, *.

**Table 1 ijms-24-09631-t001:** Baseline biochemical parameters in NAFLD and AIH patient groups. Statistical analyses were carried out by Graphpad (version 9.4.1) using Mann-Whitney test. n.s. = Not significant.

	NAFLD	(n = 24)	AIH	(n = 9)	Statistical
	Range	Mean ± SD	Range	Mean ± SD	Analysis
GOT (U/L)	23–76	44.3 ± 16.3	27–337	98.2 ± 98.9	n.s.
GPT (U/L)	12–144	65.7 ± 41.9	29–451	120.1 ± 130.6	n.s.
GGT (U/L)	8–657	132.5 ± 156.3	14–589	201.6 ± 173.4	n.s.
AP (U/L)	37–195	75.5 ± 34.6	60–158	108.1 ± 32.0	*p* = 0.0037
Albumin	4.0–5.1	4.56 ± 0.31	4.0–4.6	4.29 ± 0.26	*p* = 0.0153
Bilirubin (mg/dL)	0.17–2.51	0.55 ± 0.48	0.27–0.58	0.44 ± 0.10	n.s.

## Data Availability

All data used in this study have been included in the manuscript. The raw data generated during the current study are available from the corresponding author on reasonable request.

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
