# Peer review of "Extracellular Vesicles as Markers of Liver Function: Optimized Workflow for Biomarker Identification in Liver Disease"

_ijms, 2023, doi:10.3390/ijms24119631_

Round 1

Reviewer 1 Report

Extracellular vesicles have gathered substantial research interest in recent years, due to their potential utility as circulating biomarkers for a range of diseases, including cancer. Despite the large number of publications and the work done by the International Society for Extracellular Vesicles (ISEV) to standardize this research field, a number of problems still plague it, including (but not limited to) the lack of standardized protocols for EVs isolation.

In the present study: ‘Extracellular vesicles as markers of liver function: Optimized workflow for biomarker identification in liver disease’, by Paluschinski et al., the authors sought to circumvent the potential problem associated with isolating EVs by directly characterizing EVs in sera. They analyzed EVs directly in a complex biological context, presenting evidence showing that qualitative analysis of EVs in sera correlates with the pathological status of human patients with liver disease. In addition, the authors presented a modified PEG-based method for the separation of EVs from soluble proteins. The authors demonstrated that the enriched EVs are depleted from contaminating serum proteins, while the EVs containing fractions are enriched in exosomal markers. The authors also showed that these enriched EVs could be used as a source for identifying EVs-associated biomarkers and in the preparation of EVs for follow-up studies, including functional analysis and characterization of EVs subtypes.

Although the experiments included in this study are well introduced, logically designed, and the conclusions appear to be supported by the included data, there are a few suggestions and comments that the authors should address:

-        In the abstract/introduction section: The authors state, "In addition, PEG-enriched EVs can serve as an entry point for further downstream procedures." This statement suggests that the authors have already utilized and tested their method for this purpose. It would greatly benefit the reader if the authors could provide examples or data to support this claim. However, if the authors have not conducted such experiments yet, I recommend rephrasing the statement to indicate that the potential for PEG-enriched EVs to serve as an entry point for downstream procedures is conditional or theoretical at this stage. Furthermore, it would be crucial to address this aspect also in the discussion section.

-        In the introduction section: In general, the introduction for this kind of manuscript should be concise and straight to the point, and I find the one in this study is too long. Please focus the introduction on fewer aspects and reduce its length.

-        In the introduction section: In the introduction, the authors wrote: “no specific protein markers have been identified to distinguish between the different types of EVs”. What subtypes of EVs are the authors referring to, MVs, Exo, or? This part of the introduction is confusing please consider rephrasing it.

-        In the results section: Fig5a. In the portion of the figure representing the electropherograms of RNAs isolated from EVs-enriched pellets and EVs-depleted supernatants, the authors included cut out images of the electropherograms. To improve the clarity of the figure, please include full size electropherograms. Several studies have already published electropherograms of RNA isolated from EVs/exosomes. How do they compare with the electropherograms in your study? Please briefly discuss.

-        In the results section: In the result section the authors wrote: “hepatic stellate cells from aged rats show a senescence-associated secretory phenotype and lowered expression of matrix proteins and growth factors”. Does this refer to the preceding statement? If so please rephrase it because it is confusing, if not then please add the appropriate reference.

-        In the results section: Although the authors present an improved PEG-based method for isolating EVs, the improvements are briefly mentioned in the result section. Please include more information about the modified PEG-based method. Clarify the advantages and limitations of this method compared to existing protocols. Please provide a step-by-step description or a supplementary protocol section for the benefit of researchers attempting to replicate the study.

-        In the discussion section: The authors rightly point out the lack of standardized protocols for EV isolation as an issue in the field. To further highlight the significance of their approach, it would be beneficial if the authors could discuss how their approach compares to existing methods and why it is an improvement

- In the discussion section: The authors do not really discuss the limitation of their work. Consider expanding the discussion by discussing the limitations of your study. While you have shown that EVs in sera correlate with the disease state of human patients and animal models with liver diseases, it would be helpful to acknowledge any potential confounding factors or limitations of your approach that may affect the interpretation of your results.

Reviewer 2 Report

Martha Paluschinski et. al. investigated the extracellular vesicles in liver disease as NAFLD and AIH. They explored both the microRNAs and the cytokines in the EV and draw the conclusion that EV could be the potential biomarker for liver disease. This is a comparatively broad definition for biomarkers and the study is too primitive, as there is no specific microRNA identified for NAFLD or AIH. Meanwhile, the authors could hardly draw the conclusion that the specific microRNA in EV is only diagnosed for NAFLD or AIH, no other disease. Therefore, there are some major flaws. Here are the comments from the reviewer:

1.     There are only 9 AIH patients in this study. Too few patient samples will lead to biased conclusion.

2.     The GOT and GPT range for AIH is too large, which may result in the no significant difference between AIH patients and NAFLD patients.

3.     What does Figure 2-6 represent? The conclusion remarks for these pictures are missing from the figure legends.

4.     Also, for Figure 5, the picture had data of a, b, d, e. However, in the figure legend, there are only interpretation for Figure a, b, c. Where is Figure 5c? What does Figure 5d and 5e represents? A critical scientist should be serious on manuscript, especially the figure pictures.

5.     For Figure 5B and all the other graphs, please present the data in dots and bars together, instead of bars only.

6.     For Figure 4, why there are two alphabet “C” for Figure 4c?

7.     Please supplement scale bar for Figure 4d.

8.     What does green fluorescent represent for Figure 4d? There is no description neither on the picture nor on the figure legend.

9.     Statistics for Figure 4e is missing.

Overall, this is a poorly written manuscript with lot of flaws. The study is soundless, lacking of enough evidences to support authors’ hypothesis.  

English is fine.

Reviewer 3 Report

Reviewer comments

The present review article, ‘Extracellular vesicles as markers of liver function: Optimized workflow for biomarker identification in liver disease’ is showing the potential of extracellular vesicles as biomarkers for the diagnosis of liver disease. The present review article is very well-designed, written, and has potential results. The paper is accepted after the incorporation of the following minor corrections in the revised MS.

Comments/Suggestions

1.      Remove the full stop from the title.

2.      Line 380, p./mL.. remove the full stop after p.

3.      Line 163 and 426, the small letter used for b in Western Blot.

4.       Check the reference pattern of the journal. The journal name should be in abbreviated form.

5.      Number 1 is missing from the reference list. 

Round 2

Reviewer 2 Report

The authors carefully revised the manuscript and it could be accepted at present form.